# Toward co-design of an AI solution for detection of diarrheal pathogens in drinking water within resource-constrained contexts

**Rachel Hall-Clifford**[1]*, **Alejandro Arzu**[2], **Saul Contreras**[3], **Maria Gabriela Croissert Muguercia**[4], **Diana Ximena de Leon Figueroa**[3], **Maria Valeria Ochoa Elias**[4], **Anna Yunuen Soto Fernández**[5], **Amara Tariq**[6], **Imon Banerjee**[7,8], **Pamela Pennington**[5]

**1** Departments of Sociology and Global Health, Center for the Study of Human Health, Emory University, Decatur, GA, United States of America, **2** Center for the Study of Human Health, Emory University, Decatur, GA, United States of America, **3** Department of Computer Sciences, Universidad del Valle de Guatemala, Guatemala City, Guatemala, **4** Department of Biochemistry, Universidad del Valle de Guatemala, Guatemala City, Guatemala, **5** Center for Biotechnology Studies, Universidad del Valle de Guatemala, Guatemala City, Guatemala, **6** Machine Intelligence in Medicine and Imaging (MI-2) Lab, Mayo Clinic, Phoenix, Arizona, United States of America, **7** Department of Radiology, Mayo Clinic, Phoenix, Arizona, United States of America, **8** School of Computing and Augmented Intelligence, Arizona State University, Tempe, Arizona, United States of America

* hall-clifford@emory.edu

**Data Availability Statement:** Images generated for analysis are available in DANS repository at https://doi.org/10.17026/dans-2bd-ufvk Qualitative data

## Abstract

Despite successes on the Sustainable Development Goals for access to improved water sources and sanitation, many low and middle-income countries (LMICs) continue to struggle with high rates of diarrheal disease. In Guatemala, 98% of water sources are estimated to have E. coli contamination. This project moves toward a novel low-cost approach to bridge the gap between the microbiologic identification of *E. coli* and the vast impact that this pathogen has on human health within marginalized communities using co-designed community-based tools, low-cost technology, and AI. An agile co-design process was followed with water quality stakeholders, community staff, and local graphic design artists to develop a community water quality education mobile app. A series of alpha- and beta-testers completed interactive demonstration, feedback, and in-depth interview sessions. A microbiology lab in Guatemala developed and piloted field protocols with lay community workers to collect and process water samples. A preliminary artificial intelligence (AI) algorithm was developed to detect the presence of *E. coli* in images generated from community-derived water samples. The mobile app emerged as a pictorial and audio-driven community-facing tool. The field protocol for water sampling and testing was successfully implemented by lay community workers. Feedback from the community workers indicated both desire and ability to conduct the water sampling and testing protocol under field conditions. However, images derived from the low-cost $2 microscope in field conditions were not of a suitable quality for AI object detection of *E. coli*, and additional low-cost technologies are being considered. The preliminary AI object detection algorithm from lab-derived images performed at 94% accuracy in identifying *E. coli* in comparison to the Chromocult gold-standard.

are available in aggregate form due to the identifiable characteristics of participants. For access, contact irb@emory.edu. *Note this was a small sample of formative, qualitative interviews with local partner organization staff EcoFiltro. Due to the nature of the interviews gathering job-specific feedback and the small number of staff, the transcripts would be identifiable. The Emory IRB does not allow individual disclosure of this data.

**Funding:** Funding from the Emory Global Health Institute supported preliminary data collection (AA, MGCM, MVOE, AT). The funders had no role in study design, data collection and analysis, decision to publish, or preparation of the manuscript.

**Competing interests:** The authors have declared that no competing interests exist.

## Introduction

Despite successes on the Sustainable Development Goals for access to improved water sources and sanitation, many low and middle-income countries (LMICs) continue to struggle with high rates of diarrheal disease [1–4]. In Guatemala, an astounding 98% of water sources are estimated to have *E. coli* contamination and is a primary cause of diarrheal disease in humans [5]. As in many LMICs, poor water quality exacerbates the high rates of undernutrition and growth stunting in Guatemala, which disproportionately affect the marginalized indigenous Maya population [4, 6, 7]. In the absence of comprehensive, safe municipal water systems, marginalized populations must be empowered to understand, acquire, and regulate sources of safe drinking water to reduce transmission of waterborne diarrheal pathogens, such as *E. coli*. Two primary obstacles to accessing safe drinking water and limiting consumption of unsafe water in Guatemala are: 1) lack of education regarding the existence of microscopic pathogens invisible to the naked eye among community residents; and 2) lack of affordable and accessible technology to determine presence of pathogens by lay persons in the community.

Prior research has identified lack of education that water contamination undetectable by sight can be a source of diarrheal disease among Guatemalan community members [8]. This lack of knowledge is a critical inhibitor to behavior change, as community residents are unaware their water may be unsafe to consume [9]. Accordingly, uptake of point-of-use household water quality remediation methods, such as water filters, has been suboptimal for reducing diarrheal disease burden [10–12]. Where there is a knowledge gap that water contamination indetectable by sight can be a source of diarrheal disease, the ability to demonstrate bacterial coliform contamination in water sources has the potential to be a powerful tool for community decision-making and household behavior change. This paper describes the co-design process for the creation of a water quality educational app, development of a field protocol for water sampling and testing, and progress toward an mHealth tool to use artificial intelligence (AI)-driven object detection to identify bacterial coliforms in community water sources.

### Current water quality testing limitations

Technology appropriate for point-of-use implementation by lay end-users is imperative for improving community education, which in turn can motivate behavior change and community-level advocacy surrounding safe water. Currently, culture-based methods used for detection of *E. coli* from water are lab-dependent, time consuming (up to 3–4 days), laborious, or require advanced training that is not conducive to a low-resourced community setting. Molecular methods such as PCR require sophisticated laboratory instruments and skilled personnel. Some of the approaches used for POU water testing include biosensors that target various receptor molecules, isothermal amplification, and handheld spectrophotometer-based methods that are expensive and not practicable for community implementation, particularly in resource-constrained contexts.

This points to a need for development of a POU water quality testing mHealth device and protocol to enable early and rapid surveillance of water quality. Although mHealth tools in LMIC contexts are most frequently deployed for skilled health workers [13, 14], previous research in Guatemala has shown that mHealth tools can be successfully implemented among lay end-users when following a co-design approach [15]. Given its accessibility to lay end-users and the low-cost in comparison to laboratory equipment, we contend that the AI-driven mHealth tool could provide a key to community empowerment for improved water sources

through identification of *E. coli*-contaminated water and response at the household and community levels for increased water purification efforts and advocacy for durable safe water systems. The technological innovation in development, when combined with existing community water quality education programming, has the potential to transform clean water access in Guatemalan communities and globally.

## Guatemalan context

The absence of comprehensive water infrastructure and reliable municipal water treatment in Guatemala means that marginalized communities must navigate safe water access at the household level [2, 5]. Incomplete regulation of water and sanitation services, gaps in provision networks, and routine lapses in municipal water purification make household and community-level water quality interventions essential to reduce exposure to waterborne pathogens and, consequently, the high burden of diarrheal disease [1]. According to the WHO and UNICEF's Joint Monitoring Programme (JMP) for Water Supply and Sanitation, 5% of the Guatemalan population still consumed surface water or water from unimproved sources in 2017 [16]. The Ministry of Environmental and Natural Resources (MARN) reported that 90% of all surface water in Guatemalan rivers, lakes, and springs were contaminated to different degrees. Of approximately 24,000 controlled water systems and mechanical wells in the country, only one-third have the appropriate residue levels of chlorination (the adopted water purification method), while the remaining two-thirds presented at least some degree of unsanitary bacterial contamination [17].

According to World Bank assessments, around 40% of the rural Guatemalan population in 2014 did not obtain water from within their homes [18]. Varied systems and implementation of water purification methods at the municipal level lead to unequal access to clean water across the country [18]. Poor water quality from unimproved sources in Guatemalan communities is a regular origin of water-borne pathogens that often leads to an endemic prevalence of diarrheal disease.

Approximately 46% of Guatemalan communities have at least one household that practices open defecation [16]. Disparities in access to sanitation infrastructure persist in rural areas. For example, between 2000 and 2017, the prevalence of sewer connections in urban areas increased from 68% to 73% but decreased from 11% to 10% in rural areas. Beyond open defecation, many latrines do not properly contain waste, leading to run-off and contamination of groundwater. Basic handwashing facilities were found to be available in 77% of Guatemalan homes [16].

This research was conducted in collaboration with EcoFiltro, a Guatemalan social enterprise that produces ceramic water filters [19]. The technology of the filter and production procedures are open-source, and EcoFiltro reinvests all profits into the social arm of the business. This enables lower pricing of the basic filters for poor communities and supports the network of local community workers [11]. The company also maintains a partnership with the Ministry of Education (MINEDUC) to provide clean water in schools. Despite its successes across Guatemala, EcoFiltro data and observations from staff revealed the need to support the filter with: 1) education on the health impacts of contaminated water; 2) convincing visualization of the microbiological contamination that communities must address; and 3) a strengthened presence of EcoFiltro in communities [20]. EcoFiltro reached out to the study team with the desire for POU water quality testing, which initiated this preliminary work.

This research included: 1) co-design of a water quality app, 2) iterative development of a field protocol for water quality sampling and testing for community-based staff, and 3)

development of an AI algorithm for detection of *E. coli* bacterial coliforms. Research was conducted between July 2020 and August 2021.

## App co-design

**Methods.** We followed a co-design process for the development of an app with EcoFiltro staff to deliver educational content, including videos, on water quality and purification methods. The project was initiated by partner organization EcoFiltro as an identified quality improvement need for their community water quality education efforts. Our co-design approach facilitates end-user-led creative thinking about project goals and an iterative process of feedback on prototypes [21, 22]. This approach has been successfully used by members of the study team with other Guatemalan community-based end-users [15]. The co-design process for the app included interactive demonstrations and semi-structured individual interviews with alpha-testers and beta-testers, who were EcoFiltro staff who volunteered to participate and provided oral consent to the study team (See Table 1). The study team held weekly meetings with EcoFiltro staff to facilitate collaboration and informal feedback throughout the co-design process.

The app co-design process yielded two separate applications: 1) a version for EcoFiltro staff including the protocol for POU water quality testing (beta-tester interviews); and 2) a community version which excluded this capability (alpha-tester interviews). Interviews were coded using a grounded theory approach [23]. We created the overall structure of the app based on preliminary discussions with EcoFiltro leadership and integrated audiovisual content produced by EcoFiltro. A sequence of sprints was used to code the app using the React Native open-source mobile application framework [24, 25]. Continuous integration tests were run using a shared Git repository from GitHub, which allowed for the errors to be identified quickly, and feedback on updates from the team and EcoFiltro staff were communicated through a shared database to log updates.

**Results.** Alpha-testing interviews with EcoFiltro staff generated insights on the following key themes: 1) app revisions; 2) culturally appropriate design; 3) educational content; 4) long-term prospects of the app; and 5) the co-design process. They led to structural changes in the app, the addition of new tools and content, and modifications to existing content. There was consensus among all alpha-testers that the apps infographic content delivered on the simplicity and cultural sensitivity needed to engage with the target audience in Guatemala; one tester stated, "It is very clear, it is very specific. I love the graphic design because this is very understandable. . .for people that do not know how to read or write, and that is very important" (A3). Further suggestions for accessibility and sensitivity to community user preferences included being able to interact with the app using voice messaging instead of written messages.

**Table 1. Co-design participant demographics.**

| Demographics (n) | Alpha-tester sample (n = 6) | Beta-tester sample (n = 11) | Complete sample (n = 17) |
|---|---|---|---|
| Female, n (%) | 2 (33) | 3 (27) | 5 (29) |
| Age, median (IQR) | 34.5 (14) | 31 (6) | 31 (7) |
| Role, n (%) | | | |
| Management staff | 2 (33) | 3 (27) | 5 (29) |
| Rural field workers | 4 (67) | 5 (45) | 9 (53) |
| Urban field workers | 0 (0) | 2 (18) | 2 (12) |
| Other key informants | 0 (0) | 1 (10) | 1 (6) |
| Years at EcoFiltro, median (IQR) | 7.5 (3) | 5 (4) | 5 (4.5) |

Alpha-testers noted that the educational content in the application largely coincided with what EcoFiltro field workers already communicate to their communities, showing the compatibility of the app with work that the social enterprise already does and its potential in reducing the workload of field staff.

Alpha-testers largely agreed that the application, and more specifically the water testing capability, would increase demand for the EcoFiltro and allow for greater community engagement, an aspect of EcoFiltro's work that is lacking. Some interviews touched on how the component would provide much-needed visual evidence of bacterial contamination for communities, which was indeed the goal of the endeavor. One alpha-tester stated:

> That would help us a lot because when we are with a family, seeing the water that they consume, so they are a bit incredulous of the contamination part. Let's remember that these people's bodies, after being exposed to contaminated water, also create resistance, and when they do get sick it is at a smaller degree. For them, it is something normal. So if we have something more concrete, like something they can see, they're going to feel. . . or we're going to be able to take them to the reality that the water does indeed contain bacteria or contaminants. . . That would help us a lot. . . I find it interesting, for its application in the field (A5).

Following the alpha-tester interviews, results were coded and priority features and updates to the app were decided with EcoFiltro leadership. The study team also worked with a local graphic designer to update the design elements of the app (See Fig 1).

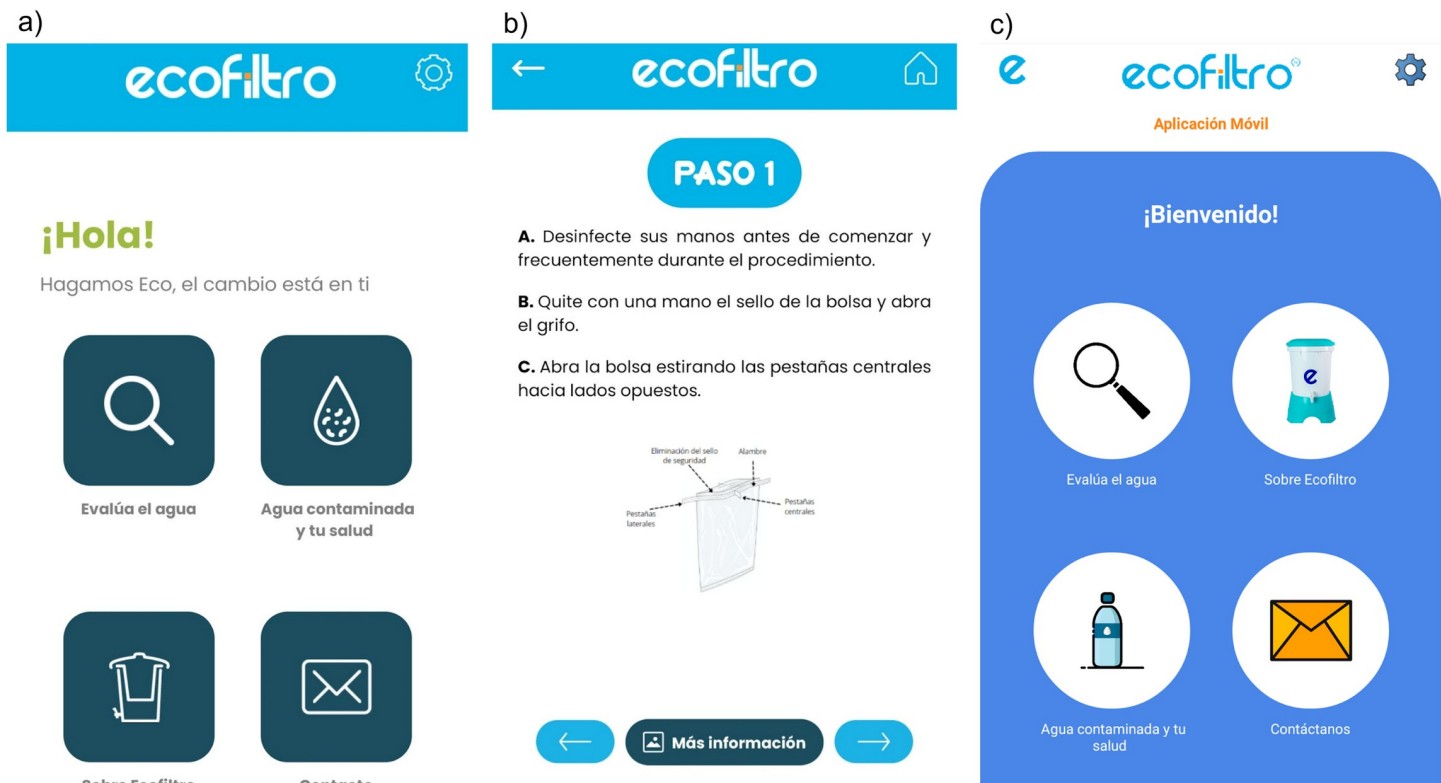

**Fig 1.** a) alpha-tester version of the app home screen; b) beta-tester version of the app home screen; c) beta-tester version of the app water sampling guidance.

The version of the app viewed by beta-testers included directions for collection, preparation, and image capture of water samples (See Fig 1C). Beta-tester interviews again generated data on the themes of app revisions, accessible design, and long-term prospects of the app. New themes arose from the beta-testers regarding the water quality testing component, including: the difficulty of the protocol, the challenges of community-based settings for performing the protocol, and training on the protocol. While all beta-testers expressed enthusiasm for having POU water quality testing capability and believed that EcoFiltro field staff could be trained to follow the field protocol, they expressed concerns about the variability of community settings in terms of cleanliness, lighting, and available facilities (e.g., water source, table) for performing the protocol. For instance, one beta-tester thought about the potential for dust to contaminate samples during the dry season:

> "I just have one question. . . for example, when one places the water on the glass slide and lets it dry, there, for example, if there was dust, do you think that dust would ruin the sample?. . . Yes it's because I see that inconvenience [occurring]. . . maybe using a box or kit [to cover the sample] or something like that where. . . or if the people or person would allow it to be performed, for example, inside a car I don't know. Right? So that there is no contamination" (B4).

Beta-testers expressed excitement to use the app and water quality testing technology, and many commented on the simplicity of the water quality testing protocol in the step-by-step guide given in the app. However, most beta-testers felt there was a need for in-depth field staff trainings on the water quality testing protocols in order to be able to implement them successfully in communities.

## Water sample field protocol development

**Methods.** In July 2020, the study team began to create the water sampling and testing protocols for community-based EcoFiltro field staff. Initial lab testing was conducted to compare images of purified water and water contaminated with *E. coli* derived from two low-cost microscopes compatible with Android phones: 1) the MagicZoom USB Microscope (adjustable magnification up to 1600X) and 2) the paper Foldscope microscope (fixed magnification of 140x), validated in previous studies [26]. Microscopic images of water samples were stored and labeled within an Emory University secure online database shared by the study team. Images were captured and stored from a variety of water sources, including: Guatemala City tap water, bottled water, surface water, and water contaminated with *E. coli* cultures. Images of samples from each of these sources were then captured before and after passing through an EcoFiltro, supporting development of the AI algorithm. During June-August of 2021, five EcoFiltro field workers were recruited as gamma-testers to participate in training sessions and provide feedback on the viability of the community-based water sampling and testing protocols, including use of the Foldscope.

**Results.** After initial field and lab-based trials using the MagicZoom USB Microscope, the team decided to use the Foldscope (See Fig 2) due to its considerably lower cost and interference from the MagicZoom light's plastic rim in resulting images. The refining of the protocols evolved around the type of consumable supplies that would be necessary to collect, prepare, and analyze water samples in a resource-constrained environment (i.e., pipettes, glass slides, staining dyes, heat sources, etc.). Initial test images using the MagicZoom revealed that crystal violet dye created microscopic images with greater texture than blue India ink; yet India ink is more widely available in shops in Guatemala. Foldscope calibration images were taken using yeast samples, which determined that heat fixation (HF) resulted in higher-quality images than air-drying (AD) as the method of fixation (See Fig 3). The degree of light exposure at the

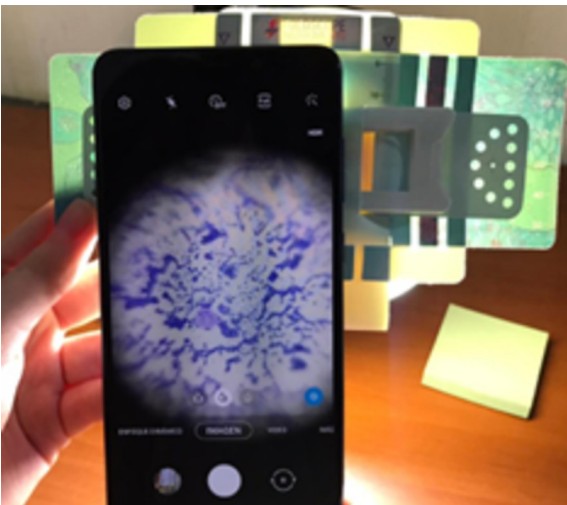

**Fig 2. The Foldscope in use.**

moment of image-capture also affected the texture and clarity of the bacteria in Foldscope images of diluted *E. coli*. Darkfield images were determined to have a clearer texture of the bacteria than brightfield images.

In June to August 2021, study team members collected water samples from community sources in the Guatemala, Sololá, Chimaltenango, and Sacatepequez Departments of Guatemala. Samples were transported to the Biochemistry Lab at the Universidad del Valle de Guatemala and processed by members of the study team within 24 hours. Two hundred images were generated from the 80 samples collected. Samples were processed using the field protocol, and 10% of the samples were tested using standard Chromocult testing (See Fig 4). Of the 8 community-derived water samples Chromocult tested, 7 were positive for *E. coli*. The images generated were used to train the AI image recognition algorithm.

In July 2021, a group of 5 EcoFiltro field staff acted as gamma-testers using the preliminary protocol for water analysis in the app. The protocol consisted of collection of community

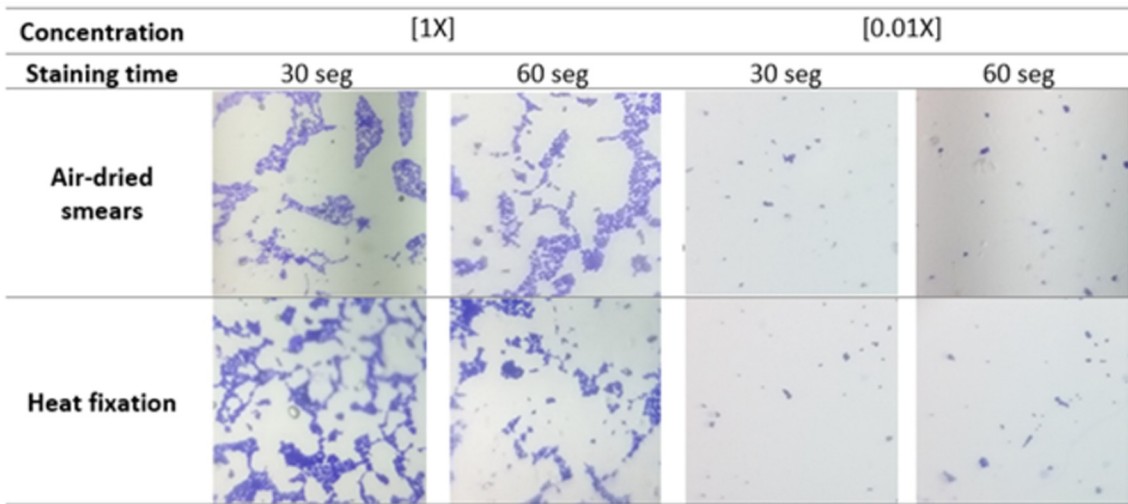

**Fig 3. Selection of slide preparation technique.**

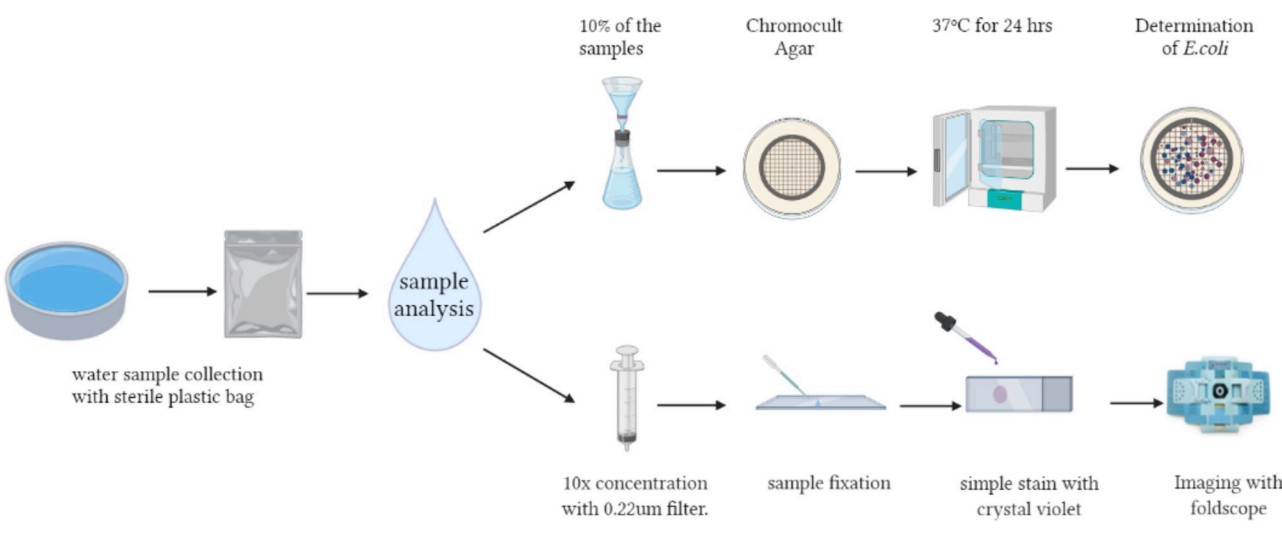

**Fig 4. Sample processing protocol.**

water-derived samples (tap, storage basin, well, or surface water sources) in sterile plastic collection bags and moving through the steps for analysis described above (syringe filtration, fixing and staining, and image capture). The gamma-testers were trained in a series of three one-hour training sessions. The training sessions covered: 1) an overview of the project and Foldscope assembly; 2) safe sample collection; and 3) slide preparation and image capture. Each training session included explanations with slides, video demonstrations, and hands-on activities. The gamma-testers each downloaded the app onto their existing Android smartphone or tablet, provided to all EcoFiltro staff. In a practical evaluation using a checklist of required steps to successfully complete the protocol, field staff completed 58% of tasks correctly, 8% with minor errors, and 34% incorrectly. From these results, we identified the need to standardize the image acquisition process since the operator's camera skills, camera app on their device, and light conditions significantly affected the results. Exit feedback with gamma-testers also indicated difficulty in mounting and using the Foldscope.

## Image recognition algorithm development

**Methods.** Using an existing database of images of water samples with *E. coli* [27] and additional images generated by the study team as described above, we developed an image analysis pipeline by experimenting with both traditional computer vision and deep learning approaches using optical microscope images on a 140X magnification. Our method uses these image analysis approaches to compare water sample images to the existing database images of *E. coli* [27] on the basis of structural morphology of *E. coli*. First, we used the shape-based techniques to detect blobs within the microscopic images of water to identify the potential candidates for pathogens. The blob detection algorithm is particularly intuitive to tag the candidate regions for pathogens as it identifies sub-pixels of an image which differ in image properties (intensity, shape) compared to surrounding regions and the properties are approximately constant within the regions. By setting up convexity and size thresholds, this algorithm was able to make near perfect distinction between dirty and clean water samples by separating them based on the number of detected blobs. In dirty water sample images, image processing engine detected pathogen presence as multiple blobs.

In contrast to the traditional computer vision, recent developments in deep learning have facilitated building of end-to-end pipelines for image processing tasks without the use of separate featurization schemes. We developed such a pipeline by using ResNet50 [28] architecture pre-trained over the generic ImageNet dataset (~3M images). The goal is for the AI object detection model to detect bacterial coliforms in water samples collected by lay end-users via comparison to a reference database [27]. Our pipeline takes microscopic images as input and classifies it as dirty or clean without requiring featurization steps such as blob detection or texture analysis. During the training phase, the model achieved >0.95 accuracy.

**Results.** We trained our image analysis models with 140x microscopic images, and our aim is to translate the models on the Foldscope (140X) and Magic Zoom (140x) images. Therefore, we performed the pre-image-quality analysis experiment with three standard biological samples: ascaris, penicillium, chlamydosporium. In our experiment, we obtained images from all the three capturing schemes in the same ambient condition (e.g., lighting): optical microscope, Foldscope, and Magic Zoom. We computed two standard metrics: 1) *structural similarity*: measure the image similarity based on luminance, structure, and contrast; and 2) *peak signal-to-noise ratio (PSNR)*: the ratio between the maximum possible power of an image and the power of corrupting noise that affects the quality of its representation, averaged over three samples. Higher value of structural similarity (maximum value of 1) and PSNR (maximum value of 48dB) indicate higher quality or better matching of image version with the baseline image. In Table 2, we present the calculated values and low value of both metrics indicating poor quality acquisition for standard samples. Translating the image analysis algorithm towards such a drop in quality is an extremely challenging problem.

Both approaches achieved near perfect results on microscopic images. While 41 blobs were detected on average in clean water samples, 311 blobs on average were found in microscopic images of *E. coli* samples indicating a clear boundary of distinction between clean and dirty samples. Similarly, the deep learning-based pipeline was able to achieve 94% accuracy on a held-out test set of 54 microscopic images (18 samples with *E. coli*). However, as expected, the model failed to generalize well for Foldscope images with 100-140x zooming and resulted in an accuracy of <65%. We have repeated the pre-image-quality analysis experiment multiple times with other pathogens and microscopic objects and obtained similar results. We concluded that such images will not be usable in their current state for pathogen detection and classification, and improvements on this limitation need to be addressed.

## Discussion

The co-design process resulted in an app that EcoFiltro is eager to deploy. The alpha- and beta-testers all expressed enthusiasm for the water quality app. The community-facing app (without water quality testing capability) was seen as an opportunity to expand the reach of and reinforce the current water quality education conducted by EcoFiltro field staff. The reach of smartphones into even poor, rural households in Guatemala is growing rapidly, and the primary way most of the population access the internet is through smartphones [29]. Harnessing this tool for providing education and opportunities for engagement with EcoFiltro could

**Table 2. Average comparative image quality metrics—microscopic compared with Foldscope and Magic Zoom.**

|  | Magic Zoom | Fold Scope |
|---|---|---|
| Structural Similarity | 0.38 | 0.31 |
| PSNR | 28.2 dB | 28.0 dB |

improve understanding of the need for water purification and uptake of household purification methods, such as water filtration.

Community education can better encourage behavior change and WASH-related advocacy when supported with real-time demonstrations of water quality. Previous studies have demonstrated that uptake of point-of-use household water quality remediation methods in Guatemala, including water filters, has been suboptimal for prevention of waterborne illnesses [10–12]. The low uptake of water quality remediation at the household and community level is related to the gap in knowledge among rural communities that pathogens invisible to the naked eye can be present in clear water that appears safe to community members. Our alpha-, beta-, and gamma-tester interviews and co-design activities indicate that this knowledge gap is a potentially highly effective area to target. This is strategy is supported by the literature. For example, an intervention in a low-resourced, rural area of Andhra Pradesh, India demonstrated that with explicit advice on specific risk reduction behaviors, individuals were more likely, by a factor of 1.5, to increase their purchase of drinking water from more reliable sources after seeing evidence that they were drinking contaminated water [30]. Further, a cross-sectional assessment of school-based WASH programs in Indonesia found that persistence of WASH-related behaviors following educational programming was significantly higher in locations with supportive infrastructure to make implementation practicable [31]. Real-time water quality testing within communities, including a visualization of the pathogen, in partnership with existing WASH education and water quality improvement projects could have significant positive impact.

The Foldscope protocol led us to a possible viable solution to detect water-borne pathogens with a low-cost technique. However, further improvements are required to guarantee the visualization of microorganisms such as *E. coli*. Our primary challenge in finalizing the field protocol for water quality sampling and testing is high-quality image capture, which has, in turn, limited real-world testing of the AI algorithm. We propose the adaptation of the Foldscope to a "fluorescent Foldscope." Fluorescence microscopy is an effective technique for real-time detection, and it is possible to use with simple mounts preparations [26]. Microorganisms (either bacteria or eukaryotic cells) are stained with a fluorescent dye, such as Syto-9, without the need for fixation, the staining procedure is simple as it involves adding the dye to the liquid sample and a 15-minute incubation at room temperature followed by the microscopic observation [32–36]. The use of Syto-9 presents some advantages, such as it is non-hazardous, has a low-cost application (around $0.05/sample) and can be transported at room temperature. For the building of the fluorescence Foldscope, we would need low-cost supplies and widely available, such as a LED light source and commercial polymeric sheets (for example Roscolux gel filters), a 3D-printed module with a borosilicate ball to be used as a condenser lens. These materials will help to build the adaptation of the Foldscope and use to visualize the samples stained with the fluorescent protocol. Although this process has not been validated, we plan to explore this option in future work because it contributes to having a rapid, simple, and field applicable method to visualize bacterial cells in water samples.

A key contributor to our challenges in high-quality image capture was the Foldscope hardware. Our next steps include working with biomedical engineers to modify the Foldscope to explore making higher resolution images, creating a fix stand to hold the device and smartphone steady, and making the Foldscope easier to attach and detach from the phone. Improvement of image quality (both still and moving) could be achieved with further refinement of camera settings in the app. We also plan to explore modification of a stand and lighting on the MagicZoom USB Microscope to generate still images and video frames for analysis as an alternate strategy [37, 38]. The success rate of our gamma-testers, lay fieldworkers with no prior lab experience, in completing the water quality sampling and testing protocol was encouraging.

After just three hours of training, they were able to collect, prepare, and fix the samples on slides. That the Foldscope attachment and image capture were the most missed steps underscores both the viability of this type of protocol and the need for improved hardware.

To refine the AI image recognition algorithm, we will develop a robust patch-based images analysis framework with deep learning to deal with image quality issues as well as noise induced due to community-based collection. The patch detection step will help us to selectively analyze the acquired images with high resolution. In addition to the patch-based analysis, we will develop a video-frame analysis method using frame-by-frame image processing of the Foldscope videos to capture the motion of the pathogens in addition to the image features. We believe that such a trajectory of analysis of the pathogens will help us to uniquely differentiate flagellated coliforms from other pathogen types as well as from other dust particles. For the video frame analysis method, we will apply a 3D CNN (3-dimensional convolutional neural network) to process the frames and understand the correlation between the frames in the temporal axis.

## Conclusion

While the challenges with hardware and image acquisition meant that we do not yet have a fully functional tool ready for broad implementation within communities, this approach shows great promise in expanding the reach of water quality testing capabilities at the community level. In addition to encouraging behavior change at the household level through education, the ability of a lay end-user to detect waterborne pathogens with a low-cost mHealth device has the potential to transform water quality monitoring and surveillance in global health. Currently, culture-based methods used for detection of *E. coli* from water are lab-dependent, time consuming, and laborious. Molecular methods such as PCR require sophisticated laboratory instruments and skilled personnel. The current approaches used for point-of-use (POU) water testing include biosensors that target various receptor molecules, isothermal amplification, and handheld spectrophotometer-based methods that are expensive and not suitable for community implementation in many LMIC contexts. This points to a need for development of a POU water quality testing mHealth device and lay field protocol to enable early and rapid surveillance of water quality.

This research moves us toward these goals. An mHealth device combining low-cost microscopy that, coupled with a smartphone, can harness AI-driven object detection to instantly detect *E. coli* coliforms in water could drastically reshape approaches to WASH education. Communities previously unaware of poor water quality will be able to see a visualization of bacterial coliforms present in community-derived water samples, which will reinforce WASH educational programming and drive uptake of community and household-level water purification methods. Further, the ability for community members to detect contaminated water can empower communities to advocate for systems level changes in availability and accessibility of clean water. Finally, we are hopeful that this technology, once developed for *E. coli*, can be trained to detect other waterborne pathogens.

## Author Contributions

**Conceptualization:** Rachel Hall-Clifford, Anna Yunuen Soto Fernández, Imon Banerjee, Pamela Pennington.

**Data curation:** Rachel Hall-Clifford, Amara Tariq.

**Formal analysis:** Rachel Hall-Clifford, Alejandro Arzu, Maria Gabriela Croissert Muguercia, Maria Valeria Ochoa Elias, Anna Yunuen Soto Fernández, Amara Tariq, Imon Banerjee.

**Funding acquisition:** Rachel Hall-Clifford.

**Investigation:** Rachel Hall-Clifford, Maria Gabriela Croissert Muguercia, Maria Valeria Ochoa Elias, Anna Yunuen Soto Fernández.

**Methodology:** Rachel Hall-Clifford, Maria Gabriela Croissert Muguercia, Maria Valeria Ochoa Elias, Imon Banerjee, Pamela Pennington.

**Project administration:** Rachel Hall-Clifford, Anna Yunuen Soto Fernández.

**Software:** Saul Contreras, Diana Ximena de Leon Figueroa.

**Supervision:** Rachel Hall-Clifford, Anna Yunuen Soto Fernández, Imon Banerjee, Pamela Pennington.

**Visualization:** Saul Contreras, Diana Ximena de Leon Figueroa, Maria Valeria Ochoa Elias, Amara Tariq.

**Writing – original draft:** Rachel Hall-Clifford, Alejandro Arzu, Anna Yunuen Soto Fernández, Amara Tariq, Imon Banerjee.

**Writing – review & editing:** Rachel Hall-Clifford, Imon Banerjee, Pamela Pennington.

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
