## [Decision Letter · Decision Letter 0]

14 Jun 2022

PGPH-D-22-00533

Toward co-design of an AI solution for detection of diarrheal pathogens in drinking water within resource-constrained contexts

Dear Rachel,

Thank you for submitting your manuscript to PLOS Global Public Health. After careful consideration, we feel that it has merit but does not fully meet PLOS Global Public Health’s publication criteria as it currently stands. Therefore, we invite you to submit a revised version of the manuscript that addresses the points raised during the review process.

We look forward to receiving your revised manuscript.

Kind regards,

Collins Otieno Asweto, PhD

Academic Editor

Journal Requirements:

1. Please update the 'Competing Interests' statement with this "The authors have declared that no competing interests exist".

3. In the online submission form, you indicated that “Data will be made available upon reasonable request. Some qualitative data are only available in aggregate form due to the identifiable characteristics of participants.”. All PLOS journals now require all data underlying the findings described in their manuscript to be freely available to other researchers, either 1. In a public repository, 2. Within the manuscript itself, or 3. Uploaded as supplementary information.

4. We do not publish any copyright or trademark symbols that usually accompany proprietary names, eg (R), (C), or TM  (e.g. next to drug or reagent names). Please remove all instances of trademark/copyright symbols throughout the text, including SytoTM.

5. Please provide separate figure files in .tif or .eps format and remove from the manuscript file.

6. Fig 2: Please confirm (a) that you are the photographer; or (b) provide written permission from the photographer to publish the photo(s) under our CC-BY 4.0 license.

Reviewers' comments:

Reviewer's Responses to Questions

**Comments to the Author**

1. Does this manuscript meet PLOS Global Public Health’s publication criteria? Is the manuscript technically sound, and do the data support the conclusions? The manuscript must describe methodologically and ethically rigorous research with conclusions that are appropriately drawn based on the data presented.

Reviewer #1: Yes

Reviewer #2: Yes

2. Has the statistical analysis been performed appropriately and rigorously?

Reviewer #1: Yes

Reviewer #2: Yes

3. Have the authors made all data underlying the findings in their manuscript fully available (please refer to the Data Availability Statement at the start of the manuscript PDF file)?

Reviewer #1: Yes

Reviewer #2: Yes

4. Is the manuscript presented in an intelligible fashion and written in standard English?

Reviewer #1: Yes

Reviewer #2: Yes

5. Review Comments to the Author

Reviewer #1: The Article meets acceptable standards. No further comments as those observed.

Reviewer #2: Reviewer’s Comment

General comments: Very innovative. When in use will support scientific intervention toward the promotion of clean water for human consumption. Thus, may reduce prevalence of E coli associate diarrhoea in low income communities

Introduction

1. The author need to provide information on:

--E. coli as a pathogenic marker of fecal coliforms and indicator of water contamination, and its consequences in causing human diseases of public health concern.

2.There is a need to highlight the operational target of the Foldscope and Magic Zoom.

For example, Cultural method focuses on isolation and morphological characteristics, Biochemical analysis target the presence of enzymes and metabolites and PCR targets the genes.

As an optical machine how does it specifically differentiate E. coli from other coliforms that may be present in the same sample.

3. Italicize “E. coli” to “E. coli”

4. How were the participants selected, what consent did they give to participate in co-designing?.

5. Reference: Format is inconsistent for example,

No 5: Chiller, T. M., Mendoza, C. E., Lopez, M. B., Alvarez, M., Hoekstra, R. M., Keswick, B. H., & Luby, S. P. (2006). Reducing diarrhoea in Guatemalan children: randomized controlled trial of flocculant-disinfectant for drinking-water. Bulletin of the World Health Organization, 84(1), 28–35

No 17: Sanchez SM, Scott K, Lopez JH. Guatemala: Closing gaps to generate more inclusive growth. World Bank; 2016.

6. PLOS authors have the option to publish the peer review history of their article (what does this mean?). If published, this will include your full peer review and any attached files.

**Do you want your identity to be public for this peer review?** For information about this choice, including consent withdrawal, please see our Privacy Policy.

Reviewer #1: **Yes: **Dr. Akude Christian

Reviewer #2: No

---

## [Editor Report · Decision Letter 1]

22 Jul 2022

Toward co-design of an AI solution for detection of diarrheal pathogens in drinking water within resource-constrained contexts

PGPH-D-22-00533R1

Dear Dr. Rachel,

We are pleased to inform you that your manuscript 'Toward co-design of an AI solution for detection of diarrheal pathogens in drinking water within resource-constrained contexts' has been provisionally accepted for publication in PLOS Global Public Health.

Best regards,

Collins Otieno Asweto, PhD

Academic Editor
